# Collaboration and Conflict—Developing Forest Restoration Techniques for Northern Thailand's Upper Watersheds Whilst Meeting the Needs of Science and Communities

**Stephen Elliott [1,\*], Sutthathorn Chairuangsri [1], Cherdsak Kuaraksa [2], Sudarat Sangkum [3], Kwankhao Sinhaseni [4], Dia Shannon [1], Phuttida Nippanon [5] and Benjapan Manohan [1]**

1   Forest Restoration Research Unit, Biology Department, Science Faculty, Chiang Mai University, Chiang Mai 50200, Thailand
2   Faculty of Technology and Community Development, Thaksin University, Chang Wat Songkhla 90000, Thailand
3   World Wildlife Fund Thailand, Phaya Thai, Bangkok 10400, Thailand
4   Environmental Future Research Institute, Griffith University, Queensland 4222, Australia
5   The Centre for People and Forests, Phahonyothin Rd., Bangkok 10900, Thailand
\*   Correspondence: stephen-elliott1@yahoo.com

**Abstract:** This paper describes an early example of Forest Landscape Restoration (FLR), which resulted from collaboration between a university, local community, and national park authority in the upper Mae Sa Valley, near Chiang Mai City, northern Thailand. Working together, the Hmong community of Ban Mae Sa Mai, Doi Suthep National Park Authority and Chiang Mai University's Forest Restoration Research Unit (FORRU-CMU) established a chronosequence of trial restoration plots from 1996 to 2013, to test the framework-species method of forest restoration. The project developed successful restoration techniques and gained insights into the factors that influence villagers' participation in forest restoration. Recovery of forest biomass, carbon storage, structure, biodiversity and ecological functioning exceeded expectations. Villagers appreciated the improved water security resulting from the project, as well as a better relationship with the park authority and increased land security. Recently, however, tree chopping and a breakdown in fire-prevention measures (perhaps symptoms of "project fatigue") have threatened the sustainability of the plot system. The project demonstrates the importance of a sound scientific basis for forest restoration projects, long-term institutional support, and appropriate funding mechanisms, to achieve sustainability.

**Keywords:** forest restoration; framework-species method

## 1. Introduction

Successful restoration of tropical forest ecosystems depends on synergies between ecological science and social science. Ecological science provides proven, effective, and practical techniques that guarantee restoration success—defined as the maximization of forest biomass, structural complexity, biodiversity and ecological functioning within climatic and edaphic limitations [1]. However, such techniques are irrelevant, unless local stakeholders have the willingness, security, skills and knowledge to use them. On the other hand, if stakeholders enthusiastically support restoration, but then implement it without regarding ecological principles (e.g., planting the wrong tree species in the wrong places at the wrong times or failing to maintain them), they become disappointed and withdraw support for further restoration initiatives.

The needs of scientific research and communities are very different. Science requires undisturbed treatment plots, control plots (where restoration is expected to fail) and meticulous data collection. The main output is original, publishable knowledge that enables practitioners to maximize the efficiency of restoration projects. In contrast, communities are more interested in using forest plots to meet their daily needs, which can disrupt experimental treatments, and they have little interest in unproductive control plots. On the other hand, local villagers often provide a willing workforce, to carry out the arduous tasks of tree planting, maintenance, monitoring and fire prevention, in return for use of the resultant forest, once the scientists' research grants have expired.

This paper describes some of the early work of Chiang Mai University's Forest Restoration Research Unit (FORRU-CMU), which adapted and refined the framework-species method of forest restoration [2] to restore tropical evergreen forest in the upper Mae Sa Valley in northern Thailand. The framework-species method involves testing tree species from the indigenous target forest ecosystem for their ability to (1) survive and grow well in deforested sites; (2) shade out weeds (with dense spreading crowns); and (3) produce resources, such as fleshy fruit or nectar-rich flowers, early in life, to attract seed-dispersing animals and consequently promote biodiversity recovery (Figure 1). This involved establishing a chronosequence of restoration plots, established annually from 1996 to 2013, in collaboration with the villagers of Ban Mae Sa Mai and Ban Mae Sa Noi, and Doi Suthep-Pui National Park Authority. The different stakeholders had different goals. FORRU-CMU's goal was to determine the most effective forest restoration techniques and the villagers wanted to strengthen their right to remain living in a national park and secure a reliable water supply for agriculture, whilst the national park authority wanted to reclaim encroached land and increase forest cover to meet national targets. Due to the age range of the plot system (5–21 years old at the time of writing), it has become an attractive site for graduate research projects on long-term tree species performance, functional traits, biodiversity recovery, and carbon sequestration. However, local support for the plot system has fluctuated over the years. Some plots have been lost to fire and encroachment. More recently, tourism development and improved road access are creating new threats to this valuable research site. In contrast, at the same time, the villagers have planted more trees, independently of the project, and organized a rally to promote fire prevention. Therefore, the socio-political situation is dynamic and complex.

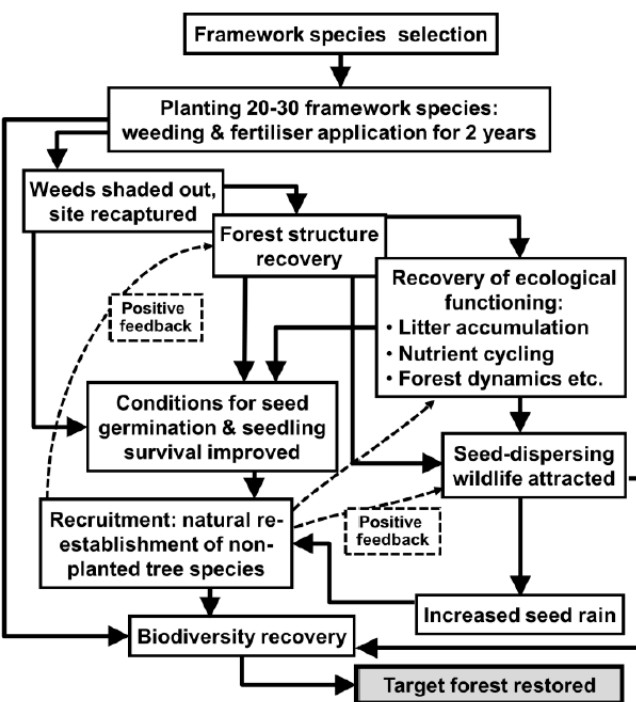

**Figure 1.** How the framework-species method works.

Although FORRU-CMU is a scientific research unit, we inevitably had to learn how to engage with local communities, as soon as we began field trials. This involved developing negotiation skills and sharing both scientific and indigenous knowledge among villagers and scientists. These activities provided FORRU-CMU with some unique insights into both the pitfalls and advantages of combining science with community needs. The purpose of this paper is to share some of these insights, so that others may emulate the positive aspects of the project, whilst avoiding the pitfalls we encountered.

## 2. Project Site, History, Structure, and Support

### 2.1. Location

The upper Mae Sa Valley lies mostly within Doi Suthep-Pui National Park (DSPNP), in Chiang Mai Province, northern Thailand, with the Hmong hill tribe communities of Ban Mae Sa Mai and Ban Mae Sa Noi (BMS) (combined population of 2197) situated at 18°52′07.24″ N, 98°51′08.47″ E, 1018 m above sea level. Additionally, the restoration trial plot system was situated at 18°51′46.62″ N, 98°50′58.81″ E, 1200–1325 m above sea level, covering 33 ha of the watershed above the village (Figure 2). The project site was assigned to FORRU-CMU by Doi Suthep-Pui National Authority in 1996, after restoration trials, established in 1995 with another Hmong community in the park (Ban Khun Chiang Kien), had failed due to re-encroachment.

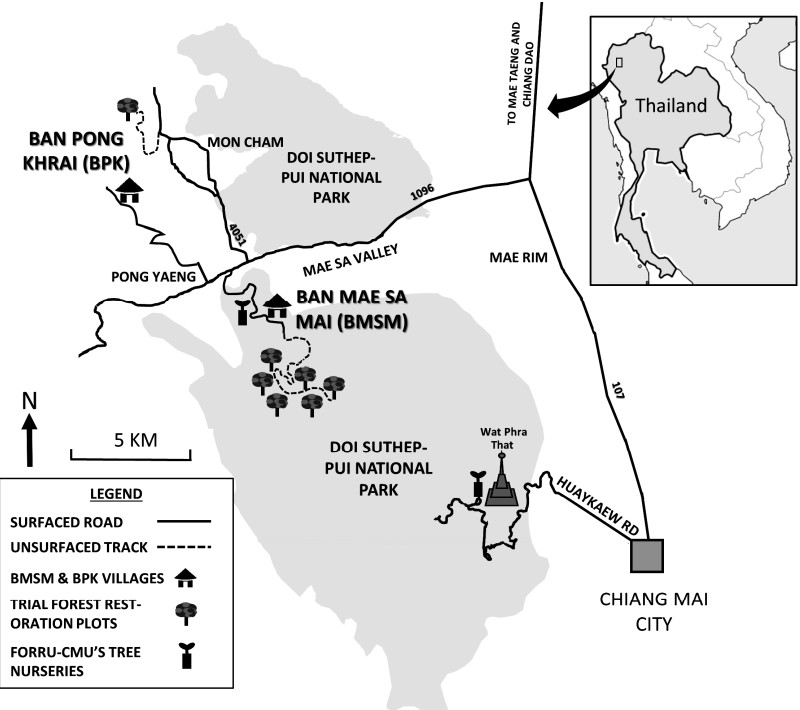

**Figure 2.** The project is situated in the upper Mae Sa Valley in northern Thailand. The grey area is the national park.

### 2.2. Soil and Climate

Bed rock was granite. Compared with soil in nearby undisturbed evergreen forest at a similar altitude, soil in the study site before planting was significantly more acidic and contained significantly less organic matter and nitrogen, more sand and less silt and clay ($p < 0.05$), which may have been due to forest clearance (Table 1).

The area has two main seasons: the wet season (May–October) and the dry season (mean monthly rainfall below 100 mm, November–April). The dry season is subdivided into the cool-dry season (November–January) and the hot-dry season (February–April). Average annual rainfall, recorded at

the weather station nearest to the study site at a similar altitude (Kog-Ma Watershed Research Station), was 1736 mm (Figure 3, [3]). Extreme temperatures ranged from a minimum of 4.5 °C in December to a maximum of 35.5 °C in March. Fire is a major constraint to reforestation in this landscape. Villagers use fire to clear land for cultivation and, despite rules to prevent accidents, fires often "escape" and burn out of control over extensive areas.

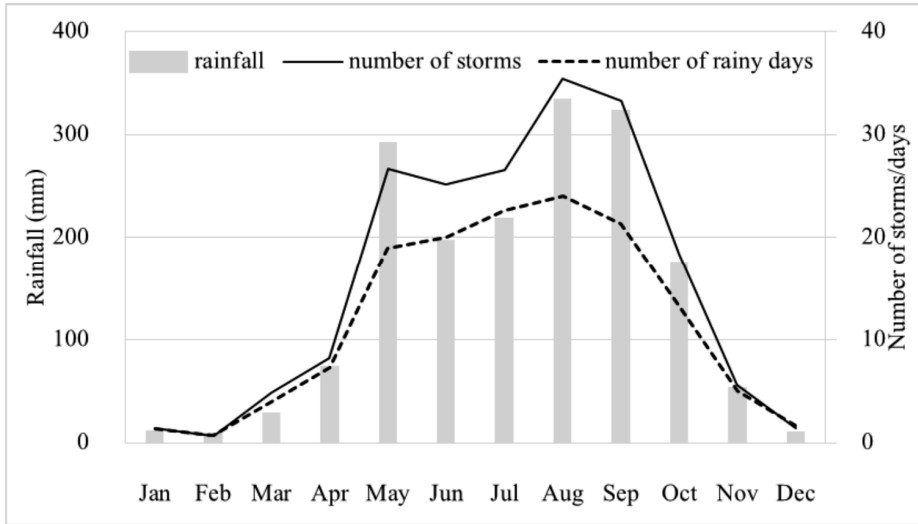

**Figure 3.** Mean monthly rainfall data for 1997–2013 (three tipping bucket automatic recording rain gauges at Huai Kog-Ma Watershed Research Station, Chiang Mai Province, 1400 m a.s.l., 9 km away from the study site) (source [3]).

**Table 1.** Soil conditions in the restoration site ($n = 16$) and nearby old-growth forest ($n = 20$) at the same altitude (source: [4]).

|  | Restoration Site | | Evergreen Forest | | |
|---|---|---|---|---|---|
|  | **Mean** | **SD** | **Mean** | **SD** | ***t*-test *p*** |
| **pH** | 5.438 | 0.423 | 6.222 | 0.545 | 0.000 |
| **Organic Matter (%)** | 5.351 | 0.997 | 7.302 | 2.480 | 0.006 |
| **Nitrogen (%)** | 0.257 | 0.045 | 0.371 | 0.121 | 0.001 |
| **Phophorus (ppm)** | 27.563 | 11.399 | 10.525 | 5.095 | 0.000 |
| **Potassium (ppm)** | 274.838 | 137.637 | 295.674 | 72.093 | 0.562 |
| **Moisture at Field Capacity (%)** | 34.755 | 2.571 | 35.345 | 4.363 | 0.636 |
| **Sand (%)** | 68.520 | 6.290 | 52.130 | 17.872 | 0.001 |
| **Silt (%)** | 18.260 | 3.090 | 22.040 | 5.473 | 0.019 |
| **Clay (%)** | 13.220 | 3.880 | 25.830 | 16.343 | 0.005 |
| **Texture** | Sandy Loam | | Sandy Loam | | na |

## 2.3. Vegetation and Stage of Degradation

Originally, the trial plot system area had been "Primary, Evergreen, Seasonal Forest" (EGF, *sensu* Maxwell & Elliott [5]) cleared from the 1950's to the early 80's to provide land for the cultivation of cabbages, potatoes, and other cash crops. The condition of the area was stage-3 degradation (*sensu* Elliott et al. [1]), Chapter 3), i.e., regenerants (remnant mature seed trees, live tree stumps capable of coppicing, tree saplings, and tree seedlings, taller than 50 cm.) at densities lower than that needed to initiate canopy closure within 2 years (<3100/ha), mostly suppressed by dominant weeds, including *Pteridium aquilinum* (L.) Kuhn (Dennstaedtiaceae); *Bidens pilosa* L. var. *minor* (Bl.) Sherf; *Ageratum conyzoides* L.; *Chromolaena odorata* (L.) R.M.King & H.Rob. and *Ageratina adenophora* (Spreng.) R.M.King & H.Rob. (all Compositae); *Commelina diffusa* Burm. F. (Commelinaceae); and grasses, e.g., *Phragmites karka* (Retz.) Trin. ex Steud. (Poaceae), *Imperata cylindrica* (L.) Raeusch., and

*Thysanolaena latifolia* (Roxb. ex Hornem.) Honda (all Poaceae). A few remnant forest trees, sparsely scattered across the plot system site, provided a potential seed source for natural forest regeneration. The nearest remnant forest, "Pah Dong Saeng", lies 2–3 km from the plots (disturbed primary EGF, regenerating following opium poppy cultivation during the 1950–60's in small patches). The villagers regard it as a *de facto* community forest and a sacred area. Potential dispersers of medium-sized seeds from that forest into the trial plots included birds (particularly bulbuls and barbets) and small mammals (civets, badgers, and small fruit bats). Dispersers of larger seeds (e.g., elephants, rhinos and wild cattle species) had been extirpated from the valley by the 1960's.

Most of the slopes below the plots were still cultivated for cabbages, with formerly extensive litchi orchards (*Litchi chinensis* Sonn. (Sapindaceae)) lower down the valley. Litchis provided the villagers with their main source of income until recently, when the over-mature litchi orchards were mostly cleared to make way for horticulture, using plastic cloches (e.g., salad vegetables, cut flowers, etc.). Over the project period, villagers invested heavily in an irrigation system that delivered piped water from the upper watershed to the agricultural field lower down the valley.

*2.4. Communities*

The community of Ban Mae Sa Mai was founded in 1922 at an altitude of about 1400 m but the village was moved down to its present location (1081 m altitude) in the early 1960's, after deforestation had caused the water supply to run dry (according to village elders). Construction of a government-funded school at that time discouraged further movement. However, the relocation event left the villagers with a strong sense of the link between deforestation and watershed services.

In 1981, the village was included within the boundaries of the newly declared Doi Suthep-Pui National Park. This meant that the villagers faced possible eviction, since they had no land titles. Furthermore, under the National Parks Act, occupation of land in a national park is illegal. To deflect possible enforcement of this law, a few villagers formed the "The Ban Mae Sa Mai Natural Resources Conservation Group" in the early 1990s, to demonstrate to authorities that they were responsible custodians of the forest. They declared Pah Dong Saeng as their community forest (even though there was no community forestry law), because it protects three springs that supply water to the village and the agricultural land below it. They also formulated a system of self-imposed penalties to deter tree-felling and hunting.

Furthermore, in 1996, the villagers decided to contribute to a national project, to celebrate His Majesty King Bhumibol Adulyadej's Golden Jubilee, which aimed to restore forest to more than 8000 km$^2$ of deforested land nationwide ("*Plook Pah Chalermphrakiat*"). They agreed to phase-out crop cultivation on 50 ha of the upper watershed and reforest the area, whilst intensifying agriculture on the more fertile land in the lower valley by installing an irrigation system. The Royal Forest Department provided them with eucalyptus and pine trees to reforest the watershed, but the villagers were disappointed with the limited species choice and poor results.

In 1996, FORRU-CMU contacted the national park authority to ask for a suitable location for field trials to test the framework-species method of forest restoration and the authority suggested that we work with the Hmong community of Ban Mae Sa Mai. When FORRU-CMU approached the village conservation group in 1996, to discuss planting framework species trial plots, they readily agreed, recognizing an opportunity to improve their previously unsuccessful efforts to reforest the 50-ha watershed area, which they had already decided to contribute to the Golden Jubilee Project.

At that time, the population of the village was about 1800. Subsequently in 2004, the community was divided into two administrative units: Ban Mae Sa Noi (current population 850) and Ban Mae Sa Mai (current population 1347) (collectively BMS in this paper). Today, the villagers' main income sources are agriculture, selling handicrafts and employment at the nearby Queen Sirikit Botanical Gardens.

## 3. Project Implementation and Monitoring

### 3.1. Initiation

The project was initiated during meetings among FORRU-CMU staff, national park officials, and key villagers, particularly leading members of the Ban Mae Sa Mae Natural Resources Conservation group, in March 1996 at the village school, with participatory sketch maps of potential restoration sites drawn on the school blackboard. Since the villagers had already committed themselves to the reforestation of 50 ha, under the Golden Jubilee Project, further social preparation was not needed. The clear vision of the conservation group was a major project asset from the start.

The committee's needs were technical support and planting stock of a wider range of tree species, both of which FORRU-CMU was well-placed to provide. During initial project meetings, it became clear that the villagers' aims were mainly political—to resolve conflict with the national park authority and deflect the threat of eviction, by demonstrating their commitment to conservation.

The park authority was under pressure to meet its commitment to reforestation targets under the Golden Jubilee Project. When the project started, FORRU-CMU was unaware of the seriousness of the conflict between the villagers and the park authority, which had deteriorated to the point of violence at that time. The unit acted as an unwitting neutral buffer between the parties. The park authority seemed happy for FORRU-CMU to take the lead in funding and organizing forest restoration in the area, so that the authority could avoid working directly with the villagers, whilst claiming credit for contributing towards the Golden Jubilee Project. Similarly, the villagers avoided direct confrontation with the park authority, by asking FORRU-CMU staff to act as a go-between.

There was no long-term plan for the project, as FORRU-CMU had funding for only 3 years at the start. Subsequently, as longer-term funding was secured incrementally, the project evolved organically, into what has since become labeled "Forest Landscape Restoration" (FLR) (although we started 7 years before the term was first coined in 2001 [6]).

### 3.2. Site Selection

Within the 50 ha that the villagers had set aside for the Golden Jubilee Project, the trial plot locations were decided upon during walks around the project area with the villagers and national park officers and by drawing sketch maps. GPS and Google Earth were used later, as those technologies became available. Practical considerations, such as ease of access and *de facto* land occupation, predominated over ecological factors in determining plot locations. Consequently, aspect, slope, land use history, etc. varied among the plots over the years. However, all plots were former evergreen forest, above 1300 m altitude, with severe forest degradation and natural regenerants absent or sparsely present at densities well below that needed to close the canopy within 3 years. Where possible, plots were established in triplicate with non-planted control plots set aside for natural regeneration since 1998, so that natural recovery of biodiversity and carbon accumulation could be compared with restoration treatment plots.

About 2 months before the start of the rainy season (March to April), plots were surveyed, using circular sample sub-plots 5 m in radius ([1], Chapter 3), to estimate the density of natural regenerants (remnant mature seed trees, live tree stumps capable of coppicing, tree saplings and tree seedlings, taller than 50 cm). The number of trees to be planted per hectare was calculated as 3100 minus the estimated density of existing natural regenerants/ha. Plots, ranging in size from 0.48 to 6.4 ha each year, were planted with 20–30 candidate framework tree species in the middle of June (4–6 weeks after the start of the rainy season), once rainfall had become reliable, since the site's altitude above the spring line precluded watering. This allowed the maximum time for root growth to access moisture lower down the soil profile before onset of the first dry season.

### 3.3. Site Preparation

Weeds were slashed down to nearly ground level about 6 weeks before planting, followed by a single application of glyphosate about 3 weeks before. This ensured weed-free conditions for about

eight weeks, which was enough time with reduced competition to facilitate establishment of the planted trees.

### 3.4. Planting Stock

At first, species choice was limited to those available from FORRU-CMU's tree nurseries. However, subsequent workshops established that the villagers had local uses for nearly all the species planted, although they showed little interest in exploiting them on a large scale. Planting stock was saplings 30–50 cm tall, grown from seeds collected locally in EGF. Seedlings were grown in plastic bags (22.8 × 6.35 cm), in a medium of forest soil, peanut husk and coconut husk in the ratio 50:25:25. Trees were initially grown in FORRU-CMU's research tree nursery in the former national park headquarters compound. However, observing the success of the initial tree-planting event in 1996, the villagers asked FORRU-CMU to sponsor the construction of a community tree nursery on the edge of the village, so that funding for planting-stock production flowed through the village economy. FORRU-CMU agreed and trained some villagers in tree propagation techniques and nursery management. Since then, FORRU-CMU has continued to employ villagers fulltime to collect seeds and grow about 20,000 trees per year (salaries currently sponsored by the Rajapruek Institute Foundation) and to pay target-related bonuses. In 2006, when one of the villagers reclaimed the original nursery site for house construction, a bigger and better nursery was built beside the main access road north of the village, on a site donated by the watershed office. Construction was sponsored by the World Wildlife Fund, with funding from King Power Duty Free.

### 3.5. Tree Planting

The villagers declared tree planting to be a community activity, which meant that every family was obliged to send at least one family member to join the activity or pay compensation to the village committee (200 THB). Plots were divided into 40 × 40 m squares: 1 "rai" in the local land-measurement system, each demarcated with poles and string. The required number of bamboo canes (to mark planting spots), saplings, and fertilizer were delivered to each rai the day before planting. Participants were divided equally among the rais, with a team leader to oversee planting within the boundaries of each. Bamboo canes were used to indicate planting spots, averaging 1.8 m between planted trees or away from natural regenerants. This spacing was initially recommended by one of the originators of the framework-species method (Tucker, pers. comm.) and was later confirmed as being optimal by a triplicated field trial in 1999, which compared the effect of average spacing of 1.5, 1.8 and 2.5 m on subsequent species recruitment [7]. Holes were dug, about twice the size of the plastic bags. Saplings were planted into the holes, and soil was added and firmed down. About 50–100 g of soluble fertilizer (NPK 15:15:15) was added in a ring, about 20–30 cm away from the base of each tree stem, and dead weeds were laid around each tree as mulch. Finally, plastic bags and any other garbage were removed from the planting sites. The species planted were varied each year to compare performance among species. Species selection for each plot/year was decided on the basis of (1) the availability of planting stock in FORRU-CMU's nurseries (with a few species obtained from forestry department nurseries), (2) performance data from previous trials (adaptive management), (3) the requirements of the experiments being performed and (4) consultation with the villagers. A stakeholders' workshop, conducted early during the project's evolution, determined that the villagers had indigenous uses for all of the species being grown in the nurseries. Useful or culturally significant species, not being grown at that time, were identified and added to subsequent field trials. Since the plots were within a national park, where economic exploitation is illegal, the economic value of species was not a primary consideration.

The basic planting protocol, described above, was also augmented each year, to test for the effects of various treatments, including spacing, fertilizer types and dosages, weeding frequency, pruning trees before planting, bare-rooted vs. containerized planting stock and the use of cardboard mulch mats [4,7].

### 3.6. Maintenance

Weeding (by hand) and fertilizer application, applied to both planted trees and natural regenerants, were repeated three times in both the first and second rainy seasons after planting, to accelerate initiation of canopy closure. To prevent fires, the villagers cut fire breaks in mid-January (at the start of the hot-dry season) and from then until mid-April (the start of the rainy season), fire prevention teams of 16 persons manned a fire look-out station in the upper watershed, 24 h a day, to detect any fires approaching the area and extinguish them. The village committee declared fire prevention to be a community activity, requiring each household to provide one family member, every 11 days, to join fire prevention patrols. At the start of each fire season, an animistic ceremony was held, to ask the village guardian spirit for successful fire prevention. If fires did not burn the planted plots, a pig was sacrificed to thank the spirit at the end of the fire season. This provided a social event, at which the villagers, FORRU-CMU staff, and national park officers could meet informally, strengthen their partnership, and plan where to plant trees in the subsequent rainy season.

### 3.7. Monitoring

Monitoring of labelled samples of the planted trees in triplicated plots (20–50 trees per species per replicate) was carried out within 2 weeks of tree planting (to provide baseline data) and was repeated at the end of the first, second, and sometimes third rainy seasons. Villagers were the primary data collectors (the committee assigned the task to the village youth club), so that they were the first to see which tree species were successful and which failed, even before the data were published.

The root collar diameter of the labelled young saplings was measured by Vernier calipers. Subsequently, as the trees grew larger, tape measures were used to record the girth at breast height (gbh) (once gbh exceeded 5 cm). Tree height was measured by tape measures and telescopic measuring poles. Simple subjective scoring was used to record tree health, weed cover (in a 1 m circle around the base of each tree), and shade (over the tree) (subjective scores on scales of 0 to 3). Data were analyzed to calculate rates of survival and growth and combined to derive performance indices. Values were compared among the species trialed each year and among the various silvicultural treatments tested (e.g., different weeding, mulching, fertilizer regimes, planting stock type, pruning, and spacing). For more details of monitoring procedures and data analysis, see [1] Chapter 7 and Appendix 2, respectively. Plots planted in 1998, 1999 and 2000 were also monitored over 6 years for age at first flowering/fruiting and attractiveness to wildlife. The oldest plots were recently resurveyed for long-term survival, growth and above-ground biomass in relation to functional traits.

The recovery of biodiversity and carbon accumulation were assessed by CMU graduate students for their thesis projects, by comparing among framework species trial plots of various ages, non-planted control plots (undergoing natural regeneration) and nearest mature forest (Pah Dong Saeng). Biodiversity studies have investigated species richness, diversity and the community composition of birds, mammals, ground flora, soil microbes, lichens and bryophytes [7–10], as well as the maintenance of genetic diversity across generations within some of the framework species planted [11–13]. Carbon studies have covered litter fall, soil carbon, and above-ground carbon in trees [14–16].

### 3.8. Benefit Sharing

The partnership between FORRU-CMU and BMS provided the project with several benefits: (1) indigenous knowledge, (2) an opportunity to test the practicability of research results with local people and (3) a supply of local labor. In addition to the environmental benefits of restoration, the partnership provided the BMS community with (1) technical expertise, (2) funding and (3) positive publicity, which helped to change the public perception of the Hmong, at that time, from forest destroyers to forest saviors. The villagers provided information on which tree species colonize abandoned fields; which ones are attractive to wildlife and which seed-dispersing animals survived

in the valley. Villagers provided their labor for all project tasks, from nursery work to planting, maintenance and monitoring of the planted trees, and fire prevention. At the end of each planting event, FORRU-CMU presented a donation to the community-development fund. These donations were mostly used to improve the irrigation system for agriculture and to upgrade roads in and around the village. FORRU-CMU paid for labor to cut the fire breaks, meals for the fire prevention teams, and for the pig sacrificed at the fire-prevention ceremony. The village committee organized teams to weed around the planted trees and apply fertilizer. FORRU-CMU paid regular daily labor rates to those who did the work. This combination of payments and voluntary inputs increased support for the project at the community level. Frequent meetings were held with the villagers, to share project tasks and particularly to decide on the positioning of the plots, so as not to conflict with existing land uses. In addition, the head of the family that had been appointed to take care of the nursery (a founding member of the conservation group), Mr. Naeng Siwapattarapong, acted as the main liaison, relaying information between FORRU-CMU staff and the village committee. As outside interest in the project grew, villagers also became involved in presenting the project to visitors and the media, thus further helping to build a positive public image of the community.

### 3.9. Capacity Building/Training/Education

Since the BMS community already had experience with tree planting and had a clear idea of their objectives (mostly political), very little additional capacity building was needed. FORRU-CMU encouraged seed collection from tree species that were not typically grown and provided training in data collection and reporting, whilst receiving much information from the villagers about the traditional uses of the trees grown for the project, as well as an increased understanding of community politics. The project has become well-known regionally and has served as a venue for numerous educational activities, from school visits to major international conferences and workshops.

### 3.10. Duration and Forms of Support

FORRU-CMU served as the main fund raiser for the project. We secured funding from a broad variety of sources, including private-sector CSR funds, from both Thai companies (e.g., Riche Monde (Bangkok) Ltd., King Power Duty Free) and foreign companies (Shell International Renewables, Guinness PLC). The Thai government contributed towards the research in the form of grants from the Biodiversity Research and Training Program. NGO's were more interested in directly supporting tree planting and the community tree nursery (WWF-Thailand, Plant a Tree Today Foundation, Rajapruek Institute Foundation). One local company even sponsored tree planting to voluntarily offset their carbon footprint (CityLife Magazine). Each grant or donation usually had a duration of 1–3 years. Therefore, once the long-term scientific value of this project became apparent, the relentless need to secure further funding to replace expiring grants was a strain. Since the restored forest was in a national park, there was no prospect of the project becoming self-sustaining financially, as the sale of products or services from the restored forest would have been illegal.

## 4. Project Outcomes and Impacts

### 4.1. Technical and Scientific Outcomes

The main technical output of the project has been an effective framework-species approach [2] for restoring EGF on stage-3 degraded land (*sensu* Elliott et al. [1]) (Figure 4). The technique involves assisted natural regeneration, complemented by planting 20–30 tree species, to rapidly restore forest biomass and structure, whilst animals, attracted to the planted trees, disperse the seeds of many other tree species into the restoration site, re-establishing biodiversity and ecosystem functioning (Figure 1).

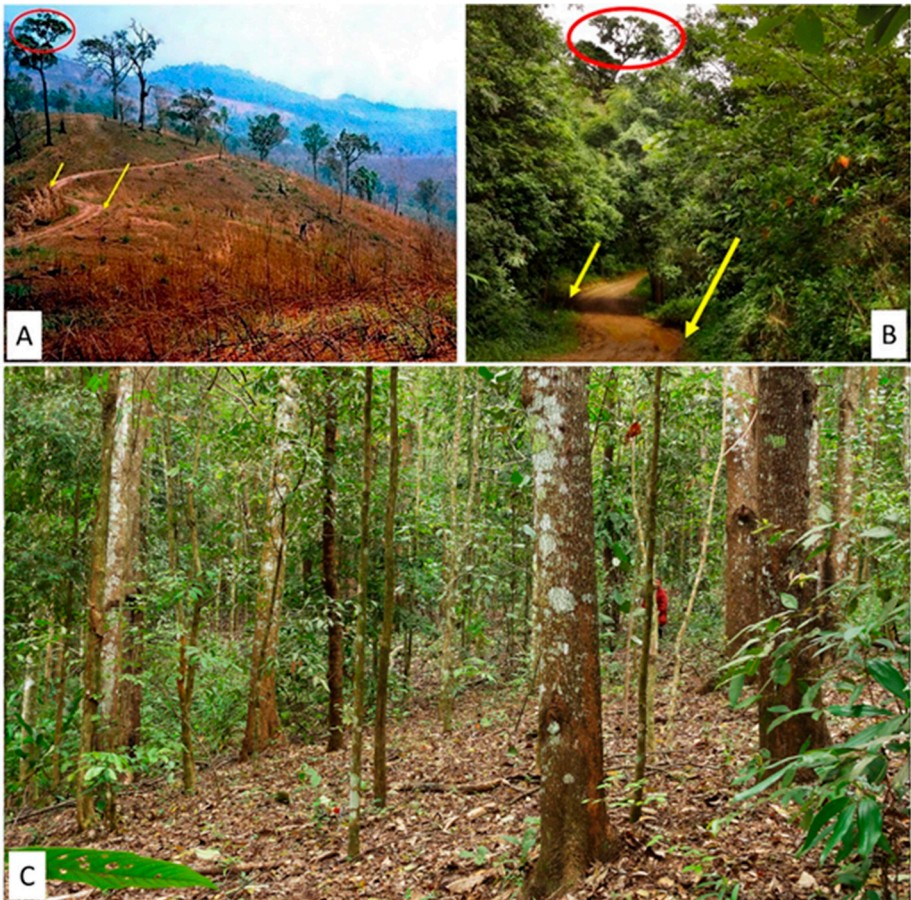

**Figure 4.** Forest restoration using the framework-species method has transformed the landscape of the upper Mae Sa Valley. (**A**) May 1998 before restoration; (**B**) same site, left of the track, restored forest, 15 years old, planted 2001; right, 9-year-old restored forest, planted 2007 (photo September 2016). (**C**) Inside nearby restored forest, $18\frac{1}{2}$ years old, a dense understory develops that comprises seedlings and saplings of >70 recruit tree species.

Candidate framework tree species (native, non-domesticated forest tree species) were selected from amongst the indigenous EGF tree flora and tested for high survival and growth rates, when planted at exposed, weedy, and deforested sites. We were looking for tree species that grow dense, broad crowns that shade out weeds, achieve rapid site recapture (the point at which planted trees overcome competition with herbaceous weeds, and canopy closure becomes inevitable) and attract seed-dispersing animals, by producing fleshy fruits at a young age or other wildlife resources (e.g., safe nesting or roosting sites). The assumption was that planted trees would encourage animals to disperse seeds from nearby forest trees into the restoration plots. Therefore, seedlings of non-planted forest tree species (termed "recruits") would establish and reinstate natural forest dynamics and succession [17].

Working with botanist J. F. Maxwell, most of the tree species, characteristic of EGF on Doi Suthep, were identified, resulting in a comprehensive herbarium collection (CMU-B Herbarium, CMU Biology Department) and a database that included habitat preferences and altitudinal distributions of >600 tree species [5]. Nursery experiments resulted in "production schedules" that detailed the most efficient treatments and timings required to propagate healthy, vigorous planting stock, of each tree species, by the optimal planting time [18]. The technical objectives of the plot system were to (1) assess the relative potential of the planted tree species to perform as framework species (survival, growth, crown expansion), (2) test the responses of the trees to various silvicultural treatments (e.g., spacing, mulching, weeding and fertilizer regimes) and (3) determine the rates of recovery of biodiversity and carbon

storage, compared with non-planted control plots (undergoing unassisted natural regeneration) and natural remnant forest.

Top-performing framework tree species were identified [19] and optimal silvicultural treatments determined, to maximize the survival and growth rates after planting [4]. With those species and treatments, canopy closure could be achieved routinely within 2–3 years after tree-planting, and biodiversity recovery was rapid. The species richness of the bird community increased from about 30 before planting, to 88 after 6 years, representing about 54% of bird species recorded in nearby mature forest [10], and the birds brought in tree seeds. Sinhaseni reported that 73 species of non-planted trees re-colonized the plot system within 8–9 years, with most having germinated from seeds dispersed from nearby forest by birds (particularly bulbuls), fruit bats, and civets [7]. The species richness of mycorrhizal fungi and lichens also increased, exceeding that of natural forest [8] and [9], respectively. Most recently, Kavinchan et al. and Jantawong et al. demonstrated remarkably rapid recovery of ecosystem carbon dynamics. Net inputs of carbon into the soil from litterfall, the overall accumulation of soil organic carbon, and the accumulation of above-ground carbon in the trees returned to levels that are typical of old-growth natural forest within 14–16, 21.5 and 16 years, respectively [14–16].

Twenty-five years ago, lack of knowledge about how to propagate, plant, and care for native forest tree species, hindered that practice of tropical forest restoration. That is no longer true for EGF in northern Thailand and for other forest types where a similar approach has been applied. Research at the BMS site established (1) which species to plant [19,20], (2) optimum seed-collection times and techniques [11,13,18], (3) optimum treatments for seed storage [21] and germination [17], (4) time and treatments needed to grow saplings to optimum size by the optimum planting time [18,22–26], (5) optimum planting techniques and spacing [7,20], (6) optimum fertilizer regimes [4] and weeding methods and frequency [17], (7) how fast biodiversity returns [7,9,10] and (8) how much carbon forest restoration can sequester [14–16] Through the project's education and outreach activities and its publications, this substantial body of knowledge has been used to improve forest restoration practices throughout SE Asia (e.g., [1,27]).

### 4.2. Socio-Economic Outcomes

From the very first village meeting, it was clear that the villagers were more interested in the potential socio-political outcomes of the project than utilitarian benefits. At that time, the villagers were in conflict with the national park authority, which had sometimes resulted in violence, and hill tribe minorities were being widely blamed, often unfairly, as the main agents of deforestation in northern Thailand. With the advent of the Golden Jubilee Project, the community recognised a chance to reduce conflict with the national park authority and deflect the threat of eviction, whilst strengthening their right of Thai citizenship, by honoring the monarchy.

From September 2005 to February 2007, structured interviews, with more than 70 community members, confirmed that villagers valued the social impacts of the project the most, followed by watershed services. About 80% of respondents said that the project had helped to reduce internal social conflicts over natural resource shortages and improved relationships between the community and authorities. Villagers also highly appreciated that the project had improved their public image. Most interviewees also stated that they had noticed an improvement in water quality, reduced soil erosion, less clogging of water pipes with silt and an increase in the reliability of the water supply during the dry season. A majority recognized that forest restoration had increased production of forest products, such as bamboo shoots and canes, banana leaves and flowers, wild vegetables and mushrooms, but such products contributed only a small amount towards household economies and were therefore of minor importance. Villagers also recognized that the project had boosted eco-tourism in the village, but the revenue from it had benefitted only a few families.

The villagers have remained living in the national park and the threat of eviction appears to have diminished. There have been no major conflicts with the park authorities and the villager's public image has improved greatly, as they have been featured in numerous TV documentaries and magazine

articles that have positively covered the project. The project has also hosted visits from the Chief of the Royal Thai Forestry Department; Britain's Minister of the Environment as well as high-level delegations from FAO, IUCN, WWF, CIFOR and RECOFTC and numerous other national and international organizations, who came to learn. This has contributed to the community's ability to leverage funds from the local government to support various infrastructure developments. Benefits have mostly been communal, increasing security for the entire village, rather than at the household level.

## 5. Challenges

### 5.1. Funding

FORRU-CMU was the sole fund-raiser for the project, securing an initial grant for 3 years' work (from Riche Monde (Bangkok) Ltd., Thailand) and finding various sponsors thereafter, which has kept the project running to date. Currently, FORRU-CMU only funds fire prevention and the village tree nursery (sponsored by Rajapruek Institute Foundation), which generates a little income from tree sales to other nearby tree-planting initiatives. The main perceived benefits from the restored forest (community security and watershed services) do not generate a cash income at the household level, whereas clearing it to grow crops would. Consequently, the long-term survival of the restored forest is at risk. Even though villagers who cleared the restored forest would run the risk of prosecution under the national park law, the park authority has a poor track record of law enforcement.

### 5.2. Land Tenure and Carbon Trading

Carbon trading could offer a means by which BMS could be compensated for agricultural production foregone, on the restored sites. However, carbon credits can only be paid to those who own the forest and are responsible for restoring or protecting it. Like thousands of villages within Thailand's protected areas, BMS exists in a legal "twilight zone". Despite the fact that the village was well-established, long before Doi Suthep-Pui National Park was gazetted, the occupation of national park land is still illegal, although the eviction of such a large community would be inhumane, impractical and politically problematic. In order for villagers to receive carbon credit payments, the government would first have to grant land rights to occupiers of national park land—a move that would most likely result in a widespread land-grab throughout Thailand's protected areas system. Furthermore, there is no a carbon-trading mechanism in Thailand, despite efforts by the Thailand Greenhouse Gases Organization (TGO) to establish one, and Thailand's REDD + program has yet to become operational, as strategy development drags on (as of May 2019). Consequently, the socio-politico conditions in Thailand currently preclude the use of carbon credits to sustain FLR, particularly within protected areas.

### 5.3. Project Fatigue and Fluctuating Community Politics

In recent years, both the numbers and aspirations of the villagers have grown, increasing pressure on the now reforested upper watershed. It is becoming apparent that some villagers may no longer be satisfied with the intangible community benefits, stated by most as their primary motivation in the 2005–2007 survey. However, more tangible benefits, such as the sale of products or environmental services from the restored site and expansion of ecotourism, are not possible under national park law. Furthermore, members of the conservation group are no longer prominent in the village committee and the current village heads do not favor conservation as much as their predecessors did.

### 5.4. Tree Felling and Re-Encroachment

Since 2010, tree felling in several plots has begun to change their structure (Figure 5A). A system of fines, previously imposed by the village committee to deter tree felling, has evidently become inoperative. Plots, selected by the village committee and national park authority for restoration, and planted in 2009 (0.64 ha), were cleared in 2012 for agriculture (Figure 5B). FORRU-CMU planted the last

experimental plots at BMS in 2013, because labor to clear a site for planting in 2014 was not forthcoming, despite having secured generous sponsorship for the work. In contrast, the villagers continued to organize tree-planting events in 2014–2016, even securing sponsorship themselves. Clearly, the community comprises both pro- and anti-restoration protagonists.

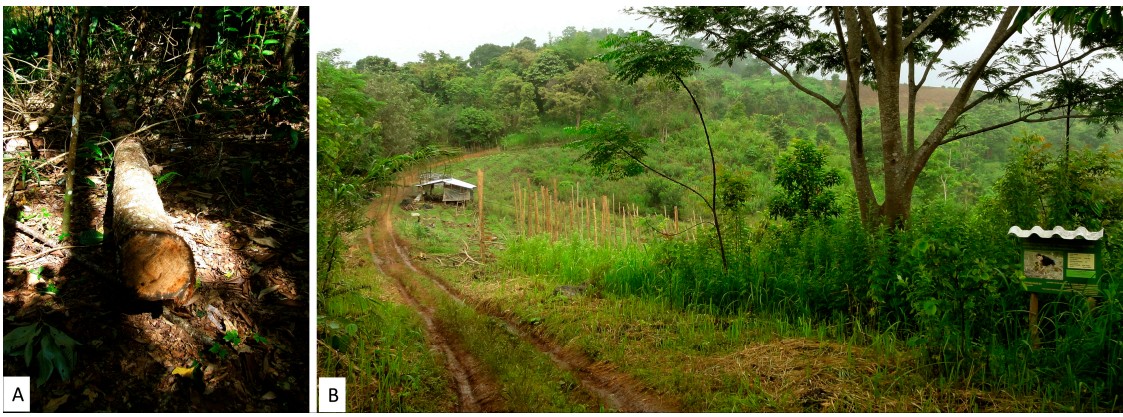

**Figure 5.** (**A**). A *Melia toosendan* tree, planted in June 1998 was chopped down in November 2010 and left to rot. (**B**). Restoration plots planted in 2009 were cleared for agriculture in 2012.

*5.5. Fire*

Undoubtedly, the greatest physical challenge has been fire. The fire-prevention system, described above, worked well until 2014. However, the villagers gradually reduced the fire prevention effort, in later years, cutting fire breaks later in the dry season and reducing the fire patrol period, even though payments for fire prevention activities were maintained (consuming around 30%–40% of the total project budget each year). There were rumors that the money paid for fire prevention was not reaching the pockets of the fire prevention workers. Consequently, the system broke down in 2015 and 2016, when approximately one third of the entire plot system was burnt. In 2015, fires cleared the understory of several plots, knocking back their regenerative potential, and in 2016, the fires were severe enough to kill mature trees in plots established in 2000. The standing dead trees now make severe fire damage more likely in the subsequent years. However, in 2018, BMS hosted 12 local Hmong communities in a high-profile ceremony to publicly recommit themselves to fire prevention. Subsequently, FORRU-CMU secured private-sector sponsorship (from GIVE and Xyliea Furniture) for a renewed, better funded fire-prevention program, with direct accountability, which has been successful so far.

*5.6. Resorts and Roads*

The villagers were able to use the project as leverage, to persuade the Royal Project to fund construction of two "ecotourism" bungalows in the village. At the same time, some families began offering homestay to visitors. These facilities were often used by the steady stream of visitors that were attracted by the project. However, a handful of families gained control over the tourism facilities and revenue in the village. Consequently, the benefits from tourism were not spread across the community. Recently, one family established a resort right on the edge of the restoration plot system. Consequently, the dirt track into the area (previously accessible only by 4WD) was recently surfaced, opening the way for further development. A second, much larger resort, has now been built nearby. If such resorts prove profitable, it is highly likely that many more will follow. Such construction is illegal in a national park.

## 6. Critical Reflections

Despite participating in village meetings, involving all households in tree planting and many in maintenance and monitoring, some families still clearly felt excluded from the project and regarded the restored forest as "belonging" to the university (in reality it belongs to the Department of National Parks,

Wildlife and Plant Conservation). Another problem was identifying and engaging key community members, as the influence of the conservation group members waned. Engagement of a social scientist on the project team from the start may have helped us to identify such problems and respond to them more effectively (e.g., by improving public participation, promoting greater understanding of the scientific aspects of the project and enabling a wider range of villagers to express their opinions about project design and implementation). Social studies, concurrent with the scientific research, would have yielded even more benefits from this project.

Our main critical reflection is that although restoration of the sub-watershed above BMS has undoubtedly transformed the landscape, "project fatigue" now threatens its long-term sustainability through encroachment, fire, tree felling, improved road access and tourism developments. Even though the villagers initiated the project themselves and FORRU-CMU provided massive financial and technical inputs over more than 20 years, such intensive support has not been enough to deter some villagers from chopping, burning and clearing some of the plots.

## 7. Lessons Learned

The main lessoned learned might be that no matter how much technical and financial support is provided, and no matter how many village meetings are run, the sustainability of FLR can never be guaranteed, if the benefits of restoration are not immediately evident and whilst rural populations continue to grow and aspirations rise. On the other hand, it does appear that the project may have had a lasting impact on the community's "conscience". The fact that the villagers have continued to plant trees on their own initiative and re-affirmed their commitment to fire prevention is hopeful.

## 8. Conclusions and Implications for Other FLR Projects

This project has established sound forest restoration principles and practices (combining protective measures with assisted natural regeneration (ANR) and tree planting) that are widely adaptable for other FLR projects throughout the seasonally dry tropics. Exponents of FLR projects often stress the need to co-operate with "the community" as if "the community" is a single entity. One of the main implications from this project is that communities comprise diverse members with diverse opinions, including those who favor forest restoration and fire prevention and those who favor agricultural expansion and the use fire both to clear fields and to score political points. Furthermore, the spectrum of opinions within communities is constantly shifting, in response to local economic and political conditions. The ever-shifting balance between restoration supporters and opponents means that working with communities will always be a rollercoaster ride, requiring constant inputs from outside, unless the economic holy grail of making standing restored forests more valuable than the crops that could replace them is achieved. Carbon credits, the marketing of watershed services, and *true* eco-tourism (where wildlife in natural habitats is the main attraction and source of income) may become viable options to do this, but only if their costs and revenues are shared equitably and transparently among all stakeholders. Changes in governance would be required to bring this about, particularly to enable park authorities, local communities and local funders to co-manage restoration projects within protected areas and share in the costs, benefits and risks involved. Furthermore, controlled, replicated landscape experiments should be established, to determine the true effects of such incentives on biodiversity and socio-economic indices and thus provide science-based inputs to co-management plans. This is vital for adaptive management—one of the central concepts of FLR.

*Addendum*

Just before going to press, at a stakeholders' meeting at BMS (16/08/19), park officials announced that the law, which controls agricultural activities in the national park, is to be more strictly enforced. Furthermore, they rejected the villagers' proposal to develop ecotourism in recompense, re-iterating that construction of tourist facilities in national parks is also illegal. FORRU-CMU provided evidence of the villagers' crucial participation in forest restoration activities since 1996. The unit also presented

satellite imagery (2005-19) of the upper watershed, showing that small shifts of field boundaries into forest areas had been more than compensated for by forest regrowth on former agricultural fields (resulting in a net increase in forest cover and a substantial increase in forest density and stature). However, it seems that the park administration is unable to take such factors into account, when enforcing the law, thus undermining the villagers' hope that contributing to forest restoration would improve their relationship with the authorities and strengthen their right to remain living in the park. Not surprisingly, therefore, some of the villagers have raised the possibility of withdrawing from fire prevention activities in the 2020 dry season. The rollercoaster ride continues.

**Author Contributions:** Conceptualization, S.E.; Methodology, all authors; Formal Analysis and Investigation, all authors; Data Curation, C.K., S.S., K.S., D.S., P.N. & B.M.; Writing-Original Draft Preparation, all authors contributed; Writing-Review & Editing, S.E, B.M.; Supervision, S.E., S.C.; Project Administration, S.E., S.C., D.S., B.M.; Funding Acquisition, S.E.

**Funding:** The sponsors of various parts of the project have been named in the main text beside, the particular activities supported.

**Acknowledgments:** It has been a great pleasure and privilege and an interesting experience to work with the villagers of Ban Mae Sa Mai/Noi, as well as Doi Suthep-Pui National Park Authority, over so many years. We thank them all. FORRU-CMU's original scientific research at BMS was sponsored by Riche Monde (Bangkok) Ltd., the Biodiversity Research and Training Program, Rajapruek Institute Foundation, Plant a Tree Today, WWF Thailand, and King Power Duty Free and Britain's Eden Project. We are very grateful for the generosity of these organizations. We also express sincere appreciation to all the very many FORRU-CMU staff and volunteers, both present and previous, who contributed to FORRU-CMU's work over so many years and to the Biology Department and Science Faculty of Chiang Mai University for continuous institutional support, since the unit's inception. Pimonrat Tiansawat and Jarik Krobtong helped with drafting the addendum.

**Conflicts of Interest:** The authors declare no conflict of interest.

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
