# Peer review of "Collaboration and Conflict—Developing Forest Restoration Techniques for Northern Thailand’s Upper Watersheds Whilst Meeting the Needs of Science and Communities"

_forests, doi:10.3390/f10090732_

Round 1
Reviewer 1 Report
General remarks
This Case Report documents a long-term study on forest restoration in northern Thailand based on the concept of Framework Species Method. Restoration in now on the global agenda and dominating forest landscape discussions. This project is in a unique position in that restoration started in 1996, and is still ongoing. Experiences gained from such projects are extremely valuable to the understanding of restoration efforts, especially when a case such as this one is reporting on a conservation conflict between indigenous communities and protected area management. A situation that could be found in many tropical regions.
I had one major issue when reading this piece. In its current state, it is difficult to tease out results that are new vs results that have been already published. Thus, I suggest to re-structure the report in a manner to increase clarity and flow of the piece. Thus, I recommend to have a more classic structure with Introduction, Methods (or Approach), Results, Discussion, which better guides the reader through the story (of course, this also depends on the journal’s rules).
Also, it would be very helpful if you could present your central concept of Framework Species Method, sketching out the main key aspects. This could be done in a Box, for example.
Specifics
Introduction: There is lots of valuable information in this section, but it needs to be re-structured and be presented differently, in more detail, in order to allow readers to get a better grasp of:
--the conceptual approach
--the field situation
--the experimental setup
--the socio-economic context = better description of the "stakeholders"
Thus, I recommend the following structure: 1) put your research in context = start on global rising of forest restoration (add a few examples from international literature; cf. below); 2) situation of forests in Thailand; flesh out conservation conflicts btw government and indigenous groups; 3) introduce the restoration approach developed by Chiang Mai University = framework species method; 4) introduce your study (key aspects) and purpose of this paper (as it is currently unclear to me why you are presenting this report).
Here some literature i suggest to use for your Introduction
White, J. C. (2019). Forests: time series to guide restoration. Nature, 567, 165-168.
Chazdon, Robin L. "Towards more effective integration of tropical forest restoration and conservation." Biotropica (2019).
Reyes‐García, Victoria, et al. "The contributions of Indigenous Peoples and local communities to ecological restoration." Restoration ecology 27.1 (2019): 3-8.
Chazdon, Robin L., et al. "Assessing the evidence base for nature's contributions to people through forest restoration and reforestation in the tropics." (2018).
Brancalion, Pedro HS, and Robin L. Chazdon. "Beyond hectares: four principles to guide reforestation in the context of tropical forest and landscape restoration." Restoration ecology 25.4 (2017): 491-496.
Thailand based literature:
Virapongse, A. "Smallholders and forest landscape restoration in upland northern Thailand." International Forestry Review 19.4 (2017): 102-119.
Pongkijvorasin, Sittidaj, and Khemarat Talerngsri Teerasuwannajak. "Win–Win Solutions for Reforestation and Maize Farming: A Case Study of Nan, Thailand." Sustainable economic development. Academic Press, 2015. 105-122.
Line 75: here starts the METHODOLOGY, which I would structure as follows: 1) describe key points of the concept Framework Species Method (can be a Box; more details than mentioned in the Introduction); 2) describe the conservation conflict between protected area mgmt. and community; 3)site selection for experimental plots based on ecological and political factors; 4) characterize the communities involved (though this could be a little paragraph under point 2);
5) monitoring of plots/ trees: what are you measuring, why, how (as many details as possible; could be summarized in a table).
Ll.76-81: a map, showing the region and field sites, and plot setup, would give much better impression than a bunch of coordinates. Thus, bring Figure 1 up. Figure 1: if existing, maybe bring community boundaries? This would show where possible overlaps exist (e.g., with the park)…if possible/ politically not “to hot” (and of course, if Communities would allow to do so/ government ditto – since I am not aware what the rules are in Thailand in this regard).
Ll.96-97: this is a table with data/ results from a different publication. You cannot use it in yours (even if some of the co-authors may be the same). Remove.
Ll.99-116: Are these your own data shown in Figure 2? Over what period/ how many years did you collect these monthly values?
L.117ff.: The subsection Vegetation and stage of degradation: very interesting information -- but text is a bit tedious to read. Why not summarize the entire paragraph in a table? This would reduce the body text and facilitate the understanding of the various degradation states.
Ll.147-148: The sentence “However, the relocation event left the…”: Unclear, please elaborate.
L.176: “participatory sketch maps”: do you mean sketch maps based on participatory mapping?
L.178: “social preparation”: unclear what you mean by this
Ll.181-183: interesting! can you elaborate more on this conflict btw park authorities and community? The fact that there was a conflict in the study area between two different stakeholders which has been ongoing for a while, and then the university came in and acted as a buffer… well that is great… this deserves ample space in your Discussion! Restoration and reforestation is not just about ecological dimension, it is mostly about people = stakeholders, land use and rights, conflicts and decision making. Your project / this report is in a fantastic position to document this situation.
L.188: Explain/ word out “FLR” as not everybody is acquainted with restoration/reforestation terminology.
L.207: what type of glyphosate? Be specific.
L235: “species planted each year were varied”: composition of plots: on what scientific base did you select which tree species to plant? how did you decide on mixing trees? (was there a consultation with the community members prior planting regarding the selection of species? Did you consider LEK (local ecological knowledge) in your restoration planning (sure you did, but emphasize this!). structure within plots: how did you decide on spacing/ placing the different species? did you randomize the trials? conditions near the plot center vs periphery may differ and have an impact on establishing probability / growth performance—how did you account for that? explain.
Ll.253-264: Monotoring: more details needed; present it a bit more technically (relate it / show clear links to the Framework Species Method. A table could do a good job here.
Ll.265-308: reads all like Discussion. Shift into DISCUSSION section.
L.309: here starts your RESULTS section. In this section, only present your own/ not yet published results/ project outcomes, else it is difficult to read/ understand.
L.317: “Candidate framework tree species”: provide a table with tree species planted in your plots; explain who selected these (university vs community vs park mgmt.); add a column on WHY each of the candidate tree species has been selected (e.g., biodiversity value, NTFP, soil value…for more WHYS, see figure on page 1 of the recent publication: Chazdon, R., & Brancalion, P. (2019). Restoring forests as a means to many ends. Science, 365(6448), 24-25.)
L.321: replace “idea” with assumption
L.353: excellent figure, especially panel A and B, very impressive! Panel C does not show much...the people here are more of a distraction than an addition
Ll.359-369: where are your own results? and here you just mention what kind of knowledge was gained during this research, but you do NOT present any results here...you just refer to Elliott et al 2013! You certainly can compare to previously published material stemming from this project, but the gain of this current contribution needs to be based on its own results/ outcomes.
L.399: here starts the DISCUSSION. Content is fine, but I would reduce the number of sub-sections. Keep Challenges and Lessons learned (the critical reflection can go into the second sub-section).
L.487ff. Main lesson: if incentives are not evident, community will not continue in conservation. If monitoring is not controlled/ guided by university or community-outside body, then established rules will be broken. Local people need to see the benefits now, not in the longer run, else, as soon as either funding dries out or “control” from the outside is loosened, commonly agreed rules are not respected any longer.
L.491: Not sure this "conscience" is helpful; people start breaking the own rules established by their older generation members…while others still continued planting trees (which is not out of guilt or to appease any outsider (maybe seen as a moral instance…). I would avoid this term.
L. 498: the community as a single entity… excellent! this is not a new finding though. See Berkes 2009 paper: Berkes, F. (2004). Rethinking community‐based conservation. Conservation biology, 18(3), 621-630.
Ll.506-509: and here lies the problem! true benefit sharing... very few working examples at global scale! so, eco-tourism is not a solution here. Rather than contemplating “the usual approaches” (carbon credits, ecotourism), it would be much more interesting / refreshing to read how you try to mitigate (reactive) or better adapt (proactive) to this sensed stakeholder fatigue? What kind of actions together with the different stakeholders could you take? Here a local solution would be beautiful. Please elaborate your reflections here.
Author Response
"Please see the attachment."

Reviewer 2 Report
I think this paper is novel and interesting for publication in “Forests”. It has a sufficient impact and it represents an undoubted advance in the subject matter, thus it is constituting a remarkable contribution to knowledge about this subject.
It is a descriptive paper of a case study of a tropical forest restoration, with its technical and social aspects. Therefore, it is not a scientific paper as such. I think a good case study.
Comments, suggestions and questions to the authors:
1) Line 85: 0:05 or = 0.05?
2) Table 1: pH
3) Line 123:"Minor" is the name of a variety and should be written in italics.
4) Line 311 and 353: What does this framework consist of? Can you explain it?
I congratulate the authors for this interesting paper.
Author Response
"Please see the attachment."

Round 2
Reviewer 1 Report
I disliked the tone of response which was aggressive and very unprofessional. Ignoring this, I think that the manuscript as is now has improved; it is more specific and detailed, with a lot of local / empirical experience/ evidence which certainly has the potential to help similar initiatives in the future.